# Is There Still a Place for Brachytherapy in the Modern Treatment of Early-Stage Oral Cancer?

**DOI:** 10.3390/cancers14010222

**Published:** 2022-01-03

**Authors:** Luboš Tuček, Milan Vošmik, Jiří Petera

**Affiliations:** 1Department of Stomatology, Faculty of Medicine in Hradec Králové and University Hospital Hradec Králové, Charles University, 500 05 Hradec Králové, Czech Republic; lubos.tucek@fnhk.cz; 2Department of Oncology and Radiotherapy, Faculty of Medicine in Hradec Králové and University Hospital Hradec Králové, Charles University, 500 05 Hradec Králové, Czech Republic; jiri.petera@fnhk.cz

**Keywords:** oral cancer, radiotherapy, brachytherapy

## Abstract

**Simple Summary:**

Brachytherapy involves the direct application of radioactive sources to the tumour. This technique is characterised by a delivery of high dose of radiation to the target volume and simultaneous sparring of healthy tissues. Historically, low-dose-rate brachytherapy played an important role in the treatment of early-stage oral cancer, with treatment outcomes that were comparable to surgery. Interest in brachytherapy as a primary treatment for oral cancer has declined in recent years due to the emergence of better surgical techniques, to advances in external beam radiotherapy, and to concerns regarding toxicity of modern high-dose-rate brachytherapy. At present, the main indications for brachytherapy are in the postoperative setting due to the superior dose conformity and better quality of life offered by brachytherapy compared to external beam radiation therapy. Postoperative brachytherapy can be administered as a monotherapy in early-stage tumours (T1N0) and in combination with elective neck dissection or EBRT to treat larger or deeper tumours. Brachytherapy yields excellent results for lip carcinoma in older patients and in tumours with unfavourable localisations. Brachytherapy is an effective salvage therapy for local recurrences in previously-irradiated areas.

**Abstract:**

Brachytherapy (BT) involves the direct application of radioactive sources to the tumour. This technique is characterised by a steep dose gradient, the delivery of high-dose radiation to the target volume centre, and the sparing of surrounding healthy tissues. Low-dose-rate (LDR) BT and manual afterloading played an important role in the treatment of early-stage oral cancer, with treatment outcomes that were comparable to surgery. Interest in BT as a primary treatment for oral cancer has declined in recent years due to the emergence of better surgical techniques, the switch from LDR BT to high-dose-rate (HDR) BT (which has a higher risk of complications), and to advances in external beam radiotherapy (EBRT). At present, the main indications for BT are in the postoperative setting due to the superior dose conformity and better quality of life offered by BT versus EBRT. Postoperative BT can be administered as monotherapy in early-stage (T1N0) cancers and in combination with elective neck dissection or EBRT to treat larger or deeper tumours. BT yields excellent results for lip carcinoma in older patients and in tumours with unfavourable localisations. BT is an effective salvage therapy for local recurrences in previously-irradiated areas. Despite its many advantages, brachytherapy is a complex treatment requiring meticulous technique and close cooperation between the radiation oncologist, physicist, and surgeon.

## 1. Introduction

Oral squamous cell carcinomas represent approximately 40% of all head and neck tumours. Smoking, alcohol consumption, and betel nut use are the major contributors in oral cancer carcinogenesis [1]. According to the GLOBOCAN 2018 report, the estimated number of new cases worldwide in 2018 was 354,846, and the number of deaths was 177.384. These data represented approximately 2% of all new cases and deaths, except non-melanoma skin tumours. The highest incidence and the leading cause of cancer death among men is in Southern Asia (mainly in India and Sri Lanka) and the Pacific Islands [2]. In Europe, between 2000 and 2007, the annual crude incidence rate was 4.78 per 100,000 people per year [3].

The main treatment modalities in early stages of head and neck squamous cell carcinomas are surgery and radiation therapy (or chemoradiation). Early stages of head and neck squamous carcinomas, including oral cavity tumours, can be treated with surgery or radiation therapy. Based on the results of retrospective studies, we suppose these modalities give similar locoregional control of the disease. According to the ESMO guidelines, the choice between surgery and radiation therapy should be based on the assessment of functional outcome and treatment morbidity for an individual patient and institutional and patient preferences [4]. In selected cases of early stages of oral cavity tumours, brachytherapy (BT) can be considered as a main treatment modality as well. The aim of this narrative review is to provide an overview of brachytherapy use as a treatment option in the early stages of squamous cell oral cancer. An electronic literature search was conducted in the PubMed and Web of Science databases for English articles published up to August 2021. Search terms used were: “oral cancer”, “early oral cancer”, “oral cancer surgery”, oral cancer radiotherapy”, “brachytherapy”, “low dose rate brachytherapy”, “high dose rate brachytherapy”, “predictive biomarkers”, and “quality of life”. The literature was selected by relevancy.

### 1.1. Brachytherapy Principles

BT is a special type of radiation based on the application of radioactive sources directly to the tumour. For many decades, BT has been successfully used to treat head and neck cancers. However, due to major advances in surgery and external beam radiotherapy (EBRT) in recent years, there is substantial uncertainty regarding the role of BT in the treatment of head and neck cancer.

BT is characterised by a steep dose gradient, excellent sparing of surrounding tissues, high doses to the tumour centre, and a short overall dose delivery time, which prevents accelerated proliferation of tumour stem cells (Figure 1). BT has several notable disadvantages, including the invasiveness of the implantation procedure and limitations in terms of the size of tumours suitable for BT (≤4 cm). Nonetheless, BT remains the most conformal form of radiotherapy, and recent studies have shown that this modality provides a better sparing of organs at risk in oral cancer than intensity-modulated radiotherapy [5].

### 1.2. Historical Background of Brachytherapy

Marie and Pierre Curie first discovered ^226^Radium in 1898, and within only a few years, radiotherapy was used in the treatment of cancer. However, several decades passed before standardised implantation and dosimetric methods were developed. In the 1930s, Paterson and Parker designed the Manchester system for interstitial radium implants [6].

BT was significantly superior to orthovoltage EBRT, which could not deliver the high doses needed to eradicate tumours due to low beam energy and acute adverse skin reactions. However, the use of radium was complicated by the risk of radioactive radon exposure and the high cost of radium sources. The popularity of BT began to wane in the early 1950s with the introduction of megavoltage cobalt machines for EBRT. In the late 1950s, BT experienced a renaissance with the discovery and manufacture of artificial radionuclides and ^192^iridium wires and seeds, which began to replace ^226^radium needles for interstitial BT. Similarly, the introduction of new afterloading implantation techniques also helped to fuel a renewed interest in BT. In the 1960s, Pierquen and Dutreix developed the Paris system of implantation and dosimetry, which remains in use today [7]. This time period—from the late 1950s through the end of the 1980s—is widely considered the golden age of brachytherapy.

Manual afterloading is a two-step process involving the implantation of inactive applicators followed by the insertion of radioactive sources. Low-dose-rate (LDR) BT (0.3–0.9 Gy/hour) was the cornerstone of brachytherapy for many years. In LDR BT, total doses range from 60 to 70 Gy administered over the course of approximately one week. In the 1990s, high-dose-rate BT (HDR BT) with automatic afterloading started to replace LDR BT (Figure 2). HDR BT has numerous advantages over LDR BT, including a better radiation safety profile, shorter treatment times, the capacity to treat more patients, and more precise dose distribution. HDR BT achieved results that were comparable to LDR BT for intracavitary treatments of gynaecological cancers and intraluminal treatment of bronchial and oesophageal stenosis. However, there were concerns about the use of HDR BT for interstitial applications due to the potentially higher risk of complications. This concern, together with the development of newer and more effective EBRT techniques, explains the decline in interest in BT in the last two decades. Another reason for this decline is that BT requires extensive, highly specialised training and experience. Nonetheless, BT continues to play an important role in several traditional indications, such as gynaecological cancers, as well as for newer clinical scenarios, particularly early-stage prostate cancer.

Brachytherapy results in better dose distribution than other treatments due to the large dose reduction in the surrounding normal tissues. Excellent local control rates and acceptable side effects have been demonstrated with BT as the sole treatment modality, postoperative method, and reirradiation method. LDR BT has been employed around the world for its superior results. With the advent of technology, HDR BT has enabled healthcare professionals to avoid radiation exposure. This therapy has been used to treat many types of cancer, such as gynaecological cancer, breast cancer, and prostate cancer. HDR BT remains an important option for oral cancer treatment [8].

### 1.3. Brachytherapy for Tongue and Floor of Mouth Cancers

In the treatment of head and neck cancer, LDR BT involved iridium hair pins or wires applied through plastic tubes (Figure 3). In oral cancer, BT was indicated as the monotherapy for early-stage (I–II) squamous cell carcinoma (SCC) and in combination with EBRT for more advanced disease and for oropharyngeal tumours. In the treatment of early-stage tongue cancer, several studies have shown that local control (LC) rates are superior when BT is a primary treatment, either as a monotherapy or combined with low-dose EBRT [9], mainly due to the ability of BT to deliver high doses of radiation in a short time. Accordingly, definitive BT was recommended for the initial treatment of stage T1N0 and T2N0 cancers. Following interstitial irradiation, most patients underwent elective neck dissection (END), with EBRT added in cases with nodal invasion. Studies published in the 1970s reported five-year LC rates for surgically treated oral cancers ranging from 81% to 85% for T1 tumours and from 77% to 85% for T2 tumours [10,11]. The 5-year LC rates achieved with LDR BT in this clinical setting (≈90%) were at least comparable to surgery [12,13] but with the additional advantage of better functional outcomes, particularly less impairment of swallowing and articulation. Compared to EBRT, BT allows for the application of higher biological doses in a shorter time with lower doses to healthy tissues, thus greatly reducing the risks of xerostomia.

In most radiotherapy departments, manual afterloading and LDR BT were largely replaced in the late 1990s by HDR BT and automatic afterloading techniques. HDR BT (dose rate ≥ 12 Gy/h) has a greater biological effect than LDR BT, and this effect is greater in late-reacting normal tissues than in cancer tissue, which explains the lower therapeutic ratio. This is why HDR BT requires a fractionated approach. In general, switching from LDR to HDR increases the likelihood of side effects for a given level of cancer control, particularly in interstitial applications. Therefore, it is important to select the optimal number of fractions and total dose of HDR BT to achieve the same effects on the tumour and normal tissues as achieved with LDR BT. Pulsed-dose-rate brachytherapy (PDR BT) was developed at the beginning of the 1990s in order to mimic the biological effect of continuous LDR BT while talking advantage of the same stepping source technology developed for HDR BT. The total dose is delivered in the same total time as with continuous LDR treatment but with a large number of small fractions (pulses)—typically one per hour and up to one per 4 h.

Numerous studies have been published on the use of LDR BT in oral cancer. By contrast, the available literature on HDR BT for this indication is much more limited (Table 1). Fractionation schedules for HDR BT typically range from 3 Gy to 6 Gy per fraction. The early results for HDR BT in oral cancer were ambiguous, with reported rates of local control (53–94%) and osteoradionecrosis (3–20%) ranging widely. Over time, as treatment centres gained more experience and recommendations regarding optimal fractionation in HDR BT were developed [14], the treatment outcomes improved significantly. The results of two randomised clinical trials in patients with oral cancer (Inoue et al. and Yamazaki et al.) showed that both techniques (LDR and HDR BT) yielded similar results [15,16]. A meta-analysis of trials comparing these two modalities in oral cancer concluded that HDR BT was comparable to LDR BT and, thus, might be suitable as a routine treatment for early-stage oral cancer [17].

### 1.4. Postoperative Brachytherapy

Due to the major advances in surgical techniques in recent years, surgery is now considered superior to BT for most indications [25]. Patients with tumours > 2 cm treated with interstitial BT alone have a particularly high risk of local recurrence. Moreover, the incidence of soft tissue necrosis and osteonecrosis increase in-line with the size of the implant volume [26]. Umeda at al. [27] compared brachytherapy (both LDR and HDR) to surgery for the treatment of stage I and II SCC of the tongue, finding that patients who underwent surgery had better local control rates, as well as better 5-year overall survival (OS) rates: 84.0% (LDR), 72.9% (HDR), and 95.4% (surgery) in stage I disease and 72.2%, 51.5%, and 93.8%, respectively, in patients with stage II disease. However, due to the risk of selection bias in that study, the results should be interpreted cautiously.

According to the latest NCCN guidelines [28], surgery is the preferred treatment for resectable tumours of the oral cavity. In this clinical setting, the functional outcomes after primary surgery are often quite good due to advances in reconstruction using microvascular techniques. In patients with positive or close margins, reresection is preferred—when feasible—over other approaches. The robotic treatment was proposed as a possible, minimally invasive treatment effective in the treatment of tumours of the oral cavity, allowing, in the case of close or positive margins, a deintensification of (chemo)radiotherapy [29]. Close resection margins (<5 mm) increase the risk of local recurrence and negatively impact survival. Inadequate margins are found in 10–16% of cases. The overall cure rate for salvage of recurrent oral spinocellular cancer is only 21–35%. In many cases, revision surgery for positive or close margins is often challenging due to reconstruction of the primary tongue defect; in these cases, adjuvant RT is an option. Postoperative radiotherapy should be considered also in cases with lymphovascular and/or perineural invasion [30]. EBRT is associated with numerous acute and late side effects, including mucositis, dysphagia, xerostomia, and neck and mandibular necrosis. The advantage of BT over EBRT in this setting is the ability to deliver highly conformal, high doses to the target in a shorter time than with EBRT while sparing the surrounding normal structures, thus avoiding the risk of xerostomia.

Several studies have assessed the role of postoperative BT in the treatment of oral cancer. Lapeyre et al. [31] evaluated LDR BT in the postoperative setting for stage T1-T2N0 oral cancer (tongue and floor of mouth) with close or positive margins, finding a LC rate of 88.5% at 2 years, which plateaued after 23 months. Grabenbauer and colleagues [26] reported a 5-year LC rate of 89% in a cohort of 37 patients treated with postoperative interstitial LDR BT alone. Pernot at al. [32] evaluated 97 patients who underwent postoperative BT for oral cavity carcinoma. The patients in that study underwent surgery followed by EBRT + BT (*n* = 51) or surgery + BT alone (*n* = 46). The 5-year LC rate was 89%, locoregional control 82%, disease-specific survival (DSS) 75%, and OS 67%. Major complications were observed in only 6% of patients.

Ianovski et al. [30] evaluated postoperative HDR BT in patients with stage T1-3N0-3M0 disease, reporting 3- and 5-year OS rates of 75.6% and 59.1%; the DSS rates were 82.3% and 68.6%, respectively. In a preliminary study carried out by our group, postoperative HDR BT was indicated in the following cases: close or positive margins, presence of lymphovascular space or perineural invasion, poor differentiation, or tumour thickness > 5 mm [33]. In that study, 30 patients received 18 x 3 Gy twice daily. The actuarial 3-year LC and OS rates were 85.4% and 73%, respectively. Nine patients (30%) developed nodal recurrence. LC was not dependent on the T-stage, probably because the tumour was surgically resected prior to BT.

### 1.5. Nodal Control

Nodal control is particularly challenging in patients with tongue cancer treated with primary or postoperative BT as a monotherapy and in patients who undergo local surgery. In stage T1-T2N0 tongue carcinoma, approximately 34% of cases present occult metastases [34]. Fujita et al. found that 36 out of 127 patients (28%) with stage T1-T2N0 tongue cancer treated by BT alone developed regional metastases, and 58% of these patients were salvaged by surgery (neck dissection) [35]. Some authors have supported salvage therapy for nodal recurrence to avoid the side effects associated with elective neck dissection and neck radiotherapy [31,36]. However, salvage therapy in neck recurrences is successful in only 50% of patients [13]. To resolve the debate surrounding the role of elective neck dissection (END), D’Cruz et al. conducted a large, randomized controlled trial to compare watchful waiting to END in patients with clinically node-negative, early-stage oral cavity cancer [37]. END was associated with significantly better OS and disease-free survival rates. The depth of invasion was a stronger predictor of nodal metastasis than the stage T1 or T2 tumour volume. The findings of that trial showed that watchful waiting was indicated only in stage T1 disease with a depth of invasion < 3 mm after treatment of the primary tumour; END was indicated in all other cases [38]. In patients with positive nodes in the pathological specimen, EBRT ± chemotherapy should be considered. Although nodal control is better in patients who receive BT + EBRT versus BT alone, local control is worse [9]. If postoperative radiotherapy is indicated, BT can be combined with elective nodal irradiation (EBRT); however, BT plus END seems to yield superior results due to the higher doses achieved with BT alone and avoids the side effects associated with EBRT. Chakrabati et al. found that BT monotherapy was associated with an 80% increased risk of late nodal metastases at a median follow-up of 65 months [39]. Those authors suggested that invasive interstitial BT may disrupt the body’s physiologic barrier, and thus, this procedure is a probable risk factor for late nodal recurrence. For this reason, they recommend prophylactic nodal irradiation in addition to BT. However, other studies have not found any association between neck failure and needle track contamination [12]. In the study by Fujita et al. [35], the incidence of nodal metastases (median follow-up, 94 months) was 28% in the BT monotherapy group versus 23% in the combination group.

### 1.6. Molecular Prognostic and Predictive Factors

A better understanding of molecular tumour biology would be extremely helpful to determine when radical surgery should be preferred to radiotherapy and when END is indicated. Almanghush et al. [40] carried out a systematic review and meta-analysis of the prognostic biomarkers for oral tongue SCC. That review included 174 studies that had evaluated 184 biomarkers, the most promising of which were cyclin D1 and vascular endothelial growth factor (VEGF). Preliminary studies conducted by our group [33,41] suggest that VEGF overexpression could be a biomarker of increased risk of neck recurrence. Recently, a 15-gene signature and prognostic nomogram was developed to help predict OS in patients with non-distant metastatic oral tongue SCC [42].

Chromosomal instability (CIN) is classically defined as an increase in the rate at which numerical or structural chromosomal aberrations are acquired in a cancer cell. The number of somatic copy number (CNA) anomalies revealed by the high-resolution genome array can be considered a surrogate marker, but several points, related to sample processing and data analysis, must be standardised. Recent works have analysed normal mucous membranes using whole-genome single nucleotide polymorphism (SNP) arrays and compared different bioinformatics tools in order to identify large somatic copy number abnormalities from the same tumour mass (double-sampled pairs) and evaluated differences in the detection of chromosomal abnormalities between distant regions of the same tumour and their influence on quantitative and qualitative analyses [43,44].

### 1.7. Lip Cancer

Brachytherapy can be used for the definitive treatment of SCC of the lip, providing excellent local control and satisfactory cosmetic results. LDR BT and HDR BT yield similar 5-year LC rates (around 95%) and 5-year OS rates ranging from 70 to 90% [45,46,47,48]. BT is mainly indicated in older patients to treat postoperative recurrences or tumours located in areas with difficult access (Figure 4). Treatment tolerance is excellent.

### 1.8. Brachyterapy for Buccal Cancer

Surgery is usually the preferred method of treating resectable buccal cancer. Vegers et al. [49] found that the results of surgery were superior over those of radiotherapy for all T and N categories, but his conclusion was based on a retrospective nonrandomized study, and it is questionable whether the surgical and radiotherapy groups were indeed comparable. Other studies have shown that radiotherapy provides comparable results as surgery but better cosmetic and functional outcomes [50,51]. Small squamous cell buccal carcinomas are ideal for brachytherapy alone or for combined BT and EBRT. This approach can be considered especially in patients with contraindications of surgery or in cases where resection would lead to unacceptable functional and/or cosmetic deficits [52]. BT combined with EBRT may also be considered in a postoperative setting in patients at high risk of recurrence.

Several BT techniques are suitable for buccal carcinoma: LDR and HDR interstitial BT, the mould technique, or 198Au or 222Rn permanent implants. The five-year LC is approximately 80% across the published studies (Table 2), with better outcomes in patients with T1 and 2 compared to the T3 and 4 stages.

### 1.9. Brachytherapy for Local Recurrence

Due to the steep dose gradients, BT is indicated for salvage therapy in limited local recurrences of oral cancer in previously irradiated areas where surgery cannot be performed. The five-year LC rates range from 17% to 71%, and the 5-year OS is around 25%, depending on the selection criteria (Table 3). However, this procedure involves a higher risk of soft tissue and bone complications compared to BT administered without prior radiotherapy.

### 1.10. Improvements of BT Techniques

Advances in surgery have improved the outcomes in early oral cancer, but on the other hand, BT as a historical rival of surgery is also constantly refining and standardising the treatment techniques. In 2017, the GEC ESTRO recommendation prioritised the use of MRI to define the tumour size, which allows for better determination of the clinical target volume and therefore reduces the risk of local recurrences after brachytherapy of squamous cell carcinomas of the head and neck [63]. In 2021, the GEC ESTRO Working Group published a proposal of dose-volume histogram constraints for adjuvant BT to minimise the risk of post-radiation osteoradionecrosis and soft tissue necrosis [64].

### 1.11. Quality of Life

Survival and tumour response parameters have traditionally been key outcome criteria in cancer studies. The current oncology clinical research takes into account not only the tumour control, overall survival, and acute and late toxicity but also the resulting quality of life (QoL) when evaluating treatment outcomes. The World Health Organization defines quality of life as “an individual’s perception of their position in life, in the context of culture and value system in their life and in relation to their goals, expectations, standards and concerns” [65].

Health-related quality of life (HRQoL) is an assessment of the effect of a disease or treatment on a patient’s well-being and daily life. A number of tools have been developed to collect and evaluate HRQoL reported by patients. Some of them are designed for patients with head and neck cancer. The European Cancer Research and Treatment (EORTC) questionnaires QLQ-C30 and QLQ-H&N35 are the most widely used. In oral cancer treatment, surgery and radiotherapy (or chemoradiotherapy) are both associated with a number of side effects that can significantly reduce a patient’s HRQoL: problems with speech, eating, swallowing, breathing, pain, gastrointestinal disorders related to olfactory or taste disorders, weight loss, xerostomia, unsatisfactory aesthetic outcome influenced their social life, etc. Patients may develop depression or anxiety. The growing importance of HRQoL has been noted in the last two decades, and a large part of clinical trials in the treatment of head and neck cancer have also addressed this topic.

Unfortunately, the number of publications on early oral cancer is limited, and there is no direct comparison of HRQoL data in surgically treated patients compared to patients treated with EBRT and patients treated with BT. Műcke et al. [66] published a comparison of HRQoL in 96 patients with early or locally advanced anterior floor of the mouth cancer treated by surgery with primary reconstruction alone or surgery and adjuvant EBRT. All patients revealed a reduced quality of life twelve months or more after treatment and reduced quality of life of a greater magnitude after EBRT, especially in the presence of osteoradionecrosis. Similarly, the combined treatment with surgery and radiotherapy caused more symptoms and poorer functioning in the study of de Graeff et al. [67].

Peisker et al. [68] evaluated the QoL in patients treated with tumour resection and immediate microvascular free lobe reconstruction using the EORTC QLQC-30 and EORTC QLQ-H&N35 questionnaires. Patients completed the questionnaires between 12 and 60 months after treatment. One hundred patients were included, and in most cases, the tumour was located in the floor of the mouth, and 67% of the tumours were in the early stages. The global QoL score was 58.3, and the average functional scale score was 76.7. The highest symptom score on QLQ-C30 in the study was for fatigue, followed by financial problems, insomnia, and pain. In the H&N35 module, the four worst symptoms were: restriction of mouth opening, dry mouth, sticky saliva, and eating in public.

Efunkoya et al. [69] reported QoL in patients after oral cancer surgical treatment using the QoL University of Washington (UW-QoL) questionnaire. The UW-QoL is a 15-item questionnaire evaluating problems related to oral function, orofacial appearance, and social interaction. A total of 68 patients with oral cancer were recruited, but only 24 patients underwent surgery and completed the postoperative QoL assessment. The overall QoL of these patients improved significantly from the time of surgery to 6 months after surgery. The mean score for “appearance” was one of the lowest domain scores, but it was given the highest domain importance. On the other hand, not all of these patients underwent a complete reconstruction at the time of their evaluation. The aesthetic appearance correlated most strongly with the highest perception of deterioration in other studies [70]. In general, the changes in QoL after surgery for oral cancer involve an initially high level of depressive symptoms and deterioration of physical functions, fatigue, appetite loss, trismus, dry mouth, sticky saliva, taste/smell, social eating, swallowing, speech, and sexuality. The negative influence of physical deterioration was counterbalanced by positive changes in emotional functioning during the first year after the treatment [67]. In patients after EBRT for head and neck cancer, although the physical, emotional, social, and cognitive functions decrease slightly, the primary complaints become xerostomia, dysphagia, appetite change, and tooth decay [71]. The neck lymph node control by laterocervical dissection can reduce the vocal outcomes, but new prosthetic opportunities improve both voice and QoL [72].

Only two studies focused on QoL after brachytherapy for early oral cancer. Yoshimura et al. [73] reported results of the QLQ-C30 and EORTC QLQ-H&N35 questionnaires in 20 patients with early oral cancer patients treated by LDR BT. The global health status was 74 at 12 months after the treatment, and none of the scores were worse than the pre-treatment values. There were no significant changes in the health status, physical function, role function, or social function. No symptoms in the QLQ-H&N35 significantly deteriorated. The scores for pain, social eating, and weight loss steadily and significantly decreased.

The QoL after HDR-BT was reported by Bajwa et al. [74]. Thirty patients were treated with EBRT and HDR BT, and two patients received HDR BT alone. Two patients died, and one patient had a disease progression during follow-up. Twenty-seven patients were available for analysis. After 2 years, none of the QLQ-C30 functional scores fell below the baseline. Pain, swallowing, sensory, speech, social contact, and social eating were worsened at 3 months but were improved within two years. Dry mouth and sticky saliva were significant determinants of QoL with delayed improvement. Almost all patients returned to normal function after treatment and reported excellent QoL after 2 years, with a global health status of 73. These BT results for QoL are very promising, although a direct comparison with surgery is not possible.

## 2. Conclusions

Surgery is considered the standard treatment for oral cancer. However, brachytherapy still has an important role to play, particularly in the postoperative setting in patients who are ineligible for revision surgery and/or in elderly patients. In mobile tongue cancer, the main challenge is nodal control, which is why brachytherapy as the sole postoperative modality is suitable only in selected cases with early-stage (T10N0) disease. In patients with stage T2 disease or tumours with a depth of invasion > 5 mm, brachytherapy should be combined with nodal dissection. Sentinel node biopsy could be the perspective approach.

Although combined treatment with EBRT and BT is acceptable in larger tumours and/or for elective nodal irradiation, local control is better with BT alone. Brachytherapy is an effective treatment modality for older patients and in tumours with unfavourable localisations. Brachytherapy can be successfully used for salvage therapy in cases with locally recurrent oral cancer in previously irradiated areas. In short, brachytherapy remains relevant today as a treatment for head and neck cancer. However, it is only appropriate in certain well-defined cases and requires a high level of training and experience, with close cooperation between the radiation oncologist, physicist, and surgeon.

## Figures and Tables

**Figure 1 cancers-14-00222-f001:**
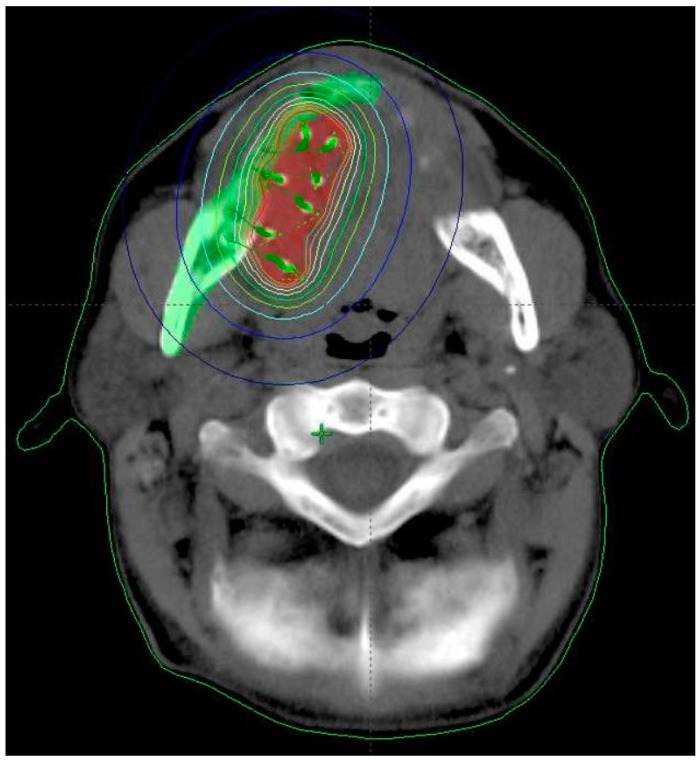
Dose distribution of brachytherapy for tongue cancer.

**Figure 2 cancers-14-00222-f002:**
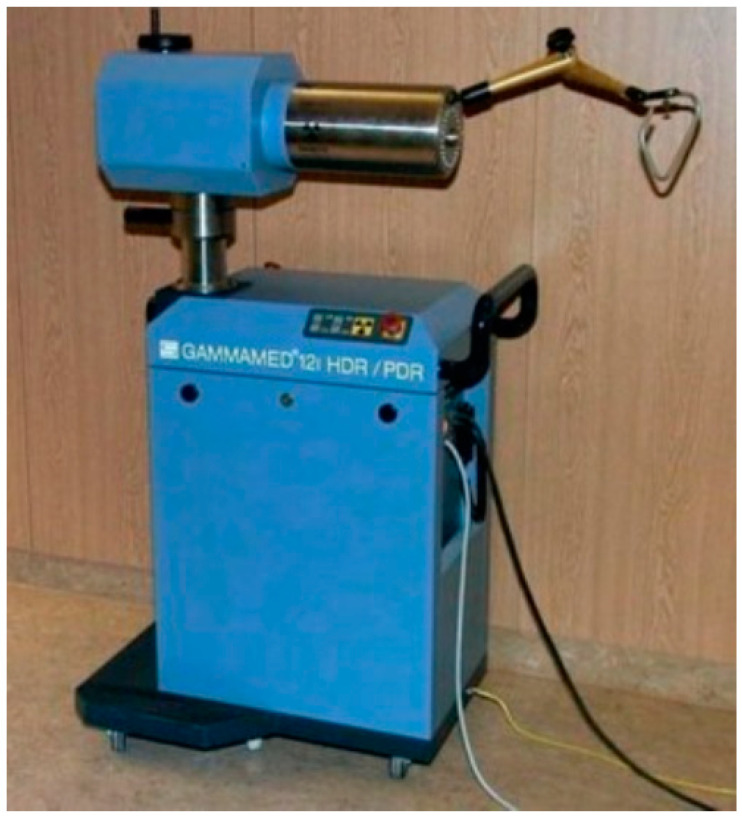
Automatic high-dose-rate brachytherapy afterloading device.

**Figure 3 cancers-14-00222-f003:**
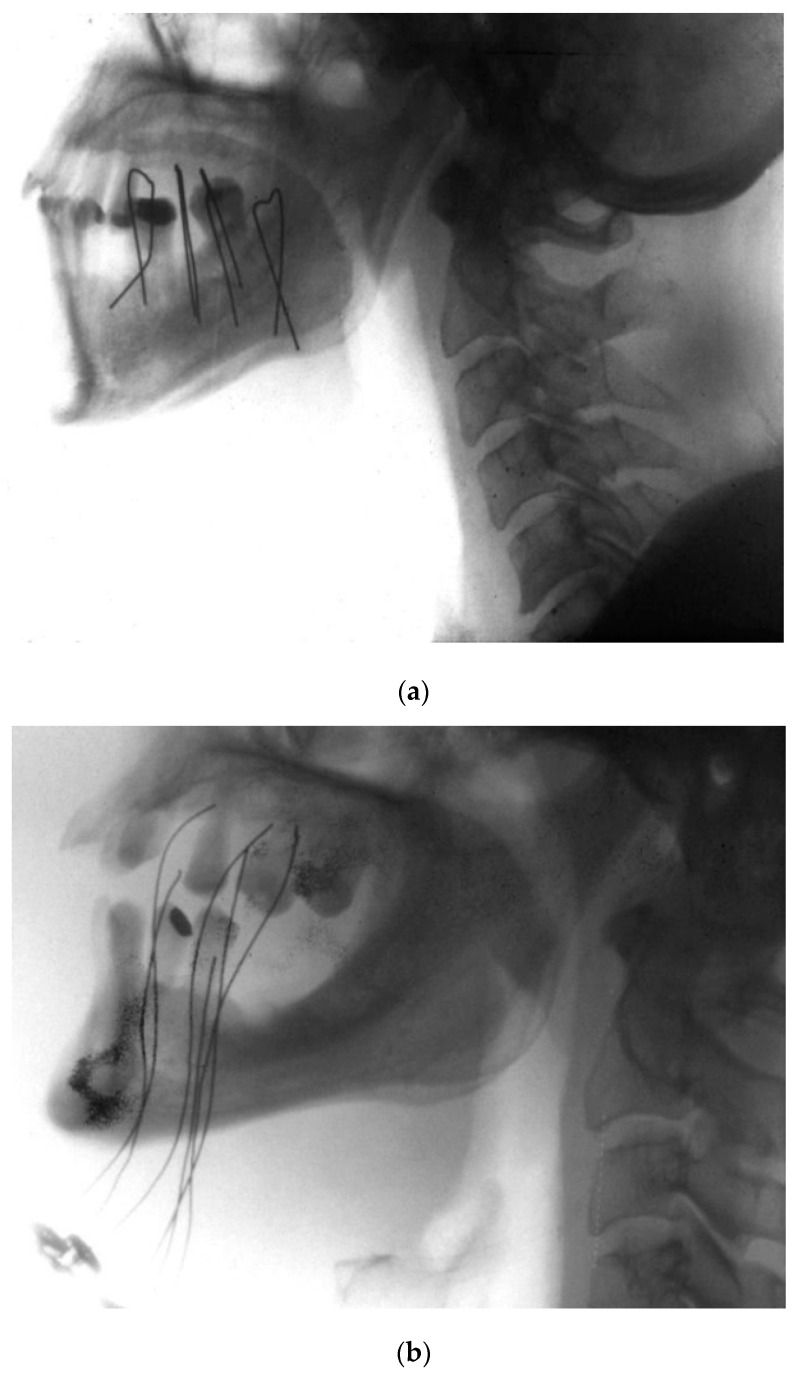
(**a**) Hair pin technique; (**b**) plastic tube technique.

**Figure 4 cancers-14-00222-f004:**
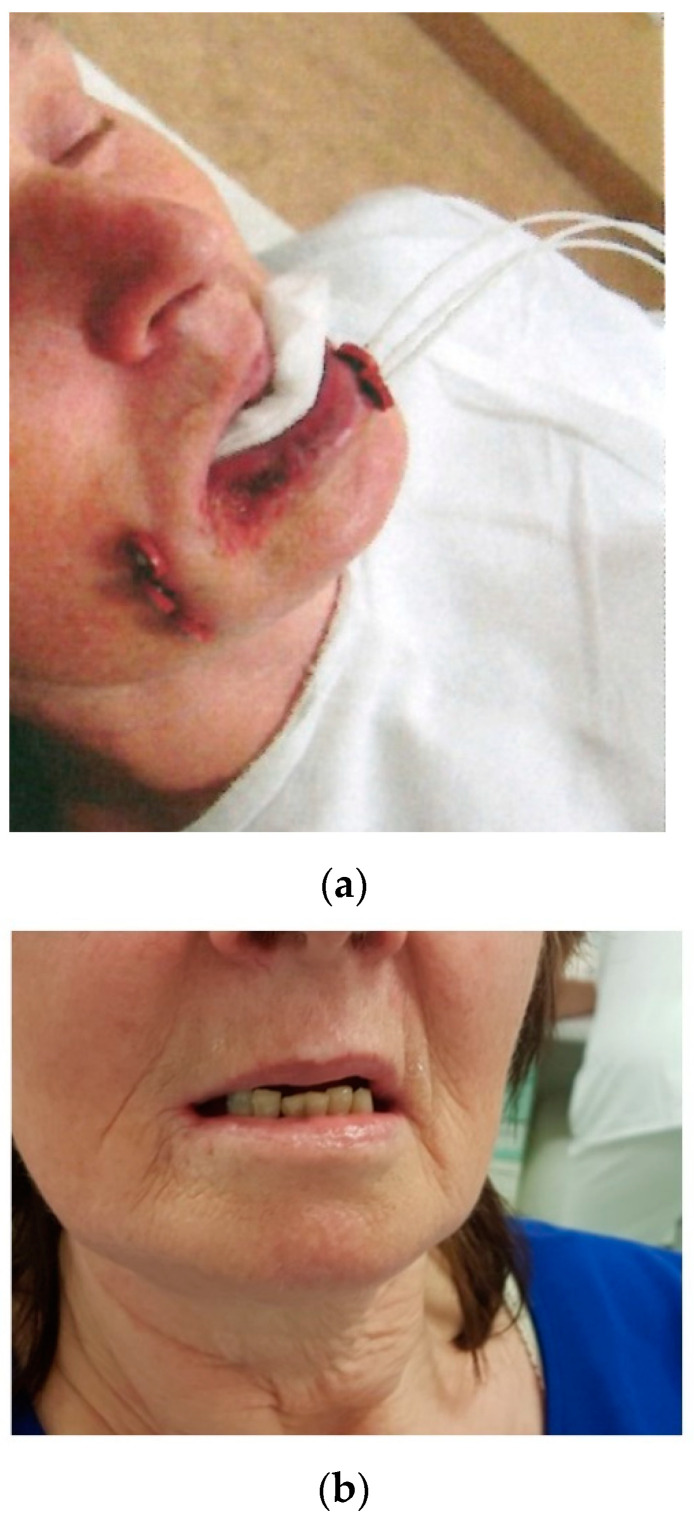
(**a**) Brachytherapy for the mouth corner. (**b**) Result after 5 years.

**Table 1 cancers-14-00222-t001:** HDR BT for stage I and II oral cancer.

Author	Year	Number of Patients	Fractionation	Results	Complications
Umeda [18]	2000	25	9–10 × 6 Gy/5 days	3y LC: 72% in stage I	Osteonecrosis: 20%
Lau [19]	1995	27	7 × 6.5 Gy	5y LC: 53%	G2 bone complications,: 11%
Inoue [15]	2001	LDR 26 HDR 25	70 Gy/4–9 days 10 × 6 Gy in 5 days	5y LC: 84% 5y LC: 87%	1 tongue ulcer 1 tongue ulcer, 2 osteoradionecrosis
Yamazaki [16]	2003	LDR 341 HDR 58	70 Gy 10 × 6 Gy in 5 days	5y LC: 85% 5y LC: 80%	bone complications: 3% and tongue ulcer: 2% in both groups
Leung [20]	2002	19	10 × 5.5 Gy in 5 days	4y LC: 94%	soft tissue + G2 bone complications: 5.2%
Guinot [21]	2010	33 pt EBRT + BT 17 pt BT only	50 Gy + 6 × 3 Gy 11 × 4 Gy	3y LC: 87% 3y LC: 100%	bone necrosis: 4% soft tissue necrosis: 16%
Matsumoto [22]	2013	67 pt BT only 35 pt EBRT + BT	10 × 5 Gy in 6 days EBRT: 20 Gy	5y LC: 94%	soft tissue necrosis: 16%
Vedasoundaram [23]	2020	15 pt BT only 11 pt BT + EBRT	11 × 3.5 Gy BT 50 Gy + 7 × 3 Gy BT	5y DFS: 100% stage I 83%: stage II 77.2%: stage III	Soft tissue necrosis: 5.6% Osteoradionecrosis: 4.8%
Bansal [24]	2016	62 pt BT alone 30 pt EBRT + BT	40–52 Gy at 4 Gy/F EBRT 40 Gy + BT 18–24 Gy at 3 Gy	5y LC: 64.2% 5y OS: 73.2% 5y DFS: 58.2% 5y nodal control rate: 83.8%	Osteoradionecrosis: 1.1% Induration, G3: 2.2%

Abbreviations: HDR, high-dose-rate; BT, brachytherapy; EBRT, external beam radiotherapy; LC, local control; OS, overall survival; DFS, disease-free survival; G, grade; y, year; pt, patients; F, fractions.

**Table 2 cancers-14-00222-t002:** Results of BT + EBRT for buccal tumours.

Author	Year	No of Patients	Technique	LC	Complications
Gerbaulet [53]	2002	266	LDR BT 65–70 Gy	81%	15–20% necrosis
Lapeyre [51]	1995	42	LDR BT 50–80 Gy	58–91%	16.7% necrosis
Vedasoundaram [54]	2014	33	HDR BT 11 × 3.5 Gy HDR BT 6 × 3.5 Gy + EBRT ± CHT	Stage I 100% Stage II 84.6% Stage III: 80%	3%
Shibuya [55]	1993	45	^198^Au or ^222^Rn average dose 81.4 Gy EBRT 20–50 Gy	88% Nodal control 77%	22% soft tissue necrosis 26% bone complications 4% both
Tayier [56]	2011	133	^198^Au ± EBRT	87%	11% soft tissue necrosis 6% osteoradionecrosis
Kotsuma [57]	2017	36	HDR BT median dose 48 Gy Mold median 15 Gy LDR T median 70 Gy ± EBRT median 30 Gy	HDR BT 82% Mold 85.7% LDR BT 72%	8% necrosis
Unetsubo [58]	2015	17	HDR BT mold 4 × 6 Gy + EBRT 30 Gy	60%	12% soft tissue necrosis

**Table 3 cancers-14-00222-t003:** Selected studies on brachytherapy for recurrent oral tumours.

Author	Patients, *n*	Tumour Site	Treatment	LC	OS
Bhalavat [59]	25	oral cavity: 15 pts oropharynx: 10 pts	HDR BT alone 40.5 Gy HDR BT 27 Gy + EBRT	2y: 75%	2y: 68%
Strnad [60]	51	oral cavity oropharynx	PDR-BT PDR-BT + EBRT ± chemotherapy	5y: 57%	5y: 26%
Bartochovska [61]	156	various	HDR BT, PDR BT	6 months: 19.6	2y: 17%
Rudžianskas [62]	30	various	HDR BT 30 Gy	2y: 67%	2y: 47%

Abbreviations: HDR, high-dose-rate; BT, brachytherapy; EBRT, external beam radiotherapy; LC, local control; OS, overall survival; PDR, pulsed dose rate; pt, patients.

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
