# Peer review of "Is There Still a Place for Brachytherapy in the Modern Treatment of Early-Stage Oral Cancer?"

_cancers, 2022, doi:10.3390/cancers14010222_

Round 1

Reviewer 1 Report

The provided literature review manuscript deals with the question of whether there is still a place for brachytherapy in the modern treatment of early-stage oral cancer? 

Following a historical overview of the different brachytherapy techniques, a narrative review of treatment outcome data is analyzed in several anatomic subsites of oral cancers.

Major remarks:

In general, since the international literature body of the topic is not very large, the reviewer believes the use of the PRISMA-type review would be more appropriate. However, the used narrative review method is also acceptable - but needs to be specified and declared in the introduction.

Line 67: Correctly: The Manchester system was first published by Paterson 1934 (BJR 7(82):592-612; BJR 7(82):612-632. The Paterson& Parker book (Ed. Meredith) was first published in 1947 in Manchester.

Line 77: The Paris system of interstitial implants was introduced by the leadership of Pirquen & Dutreix - not by Dutreix & Marinello. The respective publication needs to be listed as a reference (Pirquen et al, Acta Radiologica Oncology, 17(1978):33-48, Fasc.1)

Line 115-118: The origin of the historic radiographs needs to be specified.

Line 119: LDR and HDR are exhausting discussed - but what is with PDR? The review should be completed with this information.

Reviewer 2 Report

Minor suggestions:

  • line 40, however, the robotic treatment was proposed as a possible, minimally invasive treatment, effective in the treatment of tumors of the oral cavity, allowing in the case of close or positive margins a de-intensification of the chemoradiotherapy treatment. please discuss and cite doi: 10.1016/j.anl.2021.05.007.
  • add a flow diagram describing the study protocol, with the retrievement selection
  • line 104, Brachytherapy results in better dose distribution than other treatments due to the large dose reduction in surrounding normal tissues. Excellent local control rates and acceptable side effects have been demonstrated with brachytherapy as the sole treatment modality, postoperative method, and re-irradiation method. Low-dose brachytherapy (LDR) has been employed around the world for its superior results. With the advent of technology, high-dose brachytherapy (HDR) has enabled healthcare professionals to avoid radiation exposure. This therapy has been used to treat many types of cancer such as gynecological cancer, breast cancer and prostate cancer. HDR brachytherapy remains an important option for oral cancer treatment. please discuss and cite doi:10.1093/jrr/rrs103. 
  • line 223, Chromosomal instability (CIN) is classically defined as an increase in the rate at which numerical or structural chromosomal aberrations are acquired in a cancer cell. The number of somatic copy number (CNA) anomalies revealed by the high resolution genome array can be considered a surrogate marker, but several points, related to sample processing and data analysis, must be standardized. Recent work analyzed normal mucous membranes using whole genome SNP arrays and compared different bioinformatics tools in order to identify large somatic copy number abnormalities from the same tumor mass (double-sampled pairs) in order to evaluate differences in abnormal detection. chromosomal patterns between distant regions of the same tumor and their influence on quantitative and qualitative analysis. please discuss and cite '' S Papanikolaou V, Kyrodimos E, Tsiambas E, et al. Chromosomal instability in oral squamous cell carcinoma. J BUON. 2018;23(6):1580-1582.''  and doi:10.1016/j.cancergen.2016.11.001
  • line 322, The lymph node control of the lymph node pathology involves a laterocervical dissection which can however reduce the vocal outcomes while the conservative radiochemotherapy treatment, on the contrary, can remain in swallowing disorders and laryngotracheitis. Please discuss and cite '' Serra A, Spinato G, Spinato R, et al. Multicenter prospective crossover study on new prosthetic opportunities in post-laryngectomy voice rehabilitation. J Biol Regul Homeost Agents. 2017;31(3):803-809.''
